# Detection of *Litchi* Leaf Diseases and Insect Pests Based on Improved FCOS

**Jiaxing Xie** [1,2,3], **Xiaowei Zhang** [1], **Zeqian Liu** [1], **Fei Liao** [1], **Weixing Wang** [1,3,4] **and Jun Li** [2,5,*]

1. College of Electronic Engineering (College of Artificial Intelligence), South China Agricultural University, Guangzhou 510642, China; xjx1998@scau.edu.cn (J.X.); zhangsuancai@163.com (X.Z.); zeqianl@stu.scau.edu.cn (Z.L.); upupfei@163.com (F.L.)
2. Guangdong Laboratory for Lingnan Modern Agriculture, South China Agricultural University, Guangzhou 510642, China
3. Engineering Research Center for Monit Oring Agricultural Information of Guangdong Province, Guangzhou 510642, China
4. Zhujiang College, South China Agricultural University, Guangzhou 510900, China
5. College of Engineering, South China Agricultural University, Guangzhou 510642, China
* Correspondence: autojunli@scau.edu.cn

**Abstract:** *Litchi* leaf diseases and pests can lead to issues such as a decreased *Litchi* yield, reduced fruit quality, and decreased farmer income. In this study, we aimed to explore a real-time and accurate method for identifying *Litchi* leaf diseases and pests. We selected three different orchards for field investigation and identified five common *Litchi* leaf diseases and pests (*Litchi* leaf mite, *Litchi* sooty mold, *Litchi* anthracnose, *Mayetiola* sp., and *Litchi* algal spot) as our research objects. Finally, we proposed an improved fully convolutional one-stage object detection (FCOS) network for *Litchi* leaf disease and pest detection, called FCOS for Litch (FCOS-FL). The proposed method employs G-GhostNet-3.2 as the backbone network to achieve a model that is lightweight. The central moment pooling attention (CMPA) mechanism is introduced to enhance the features of *Litchi* leaf diseases and pests. In addition, the center sampling and center loss of the model are improved by utilizing the width and height information of the real target, which effectively improves the model's generalization performance. We propose an improved localization loss function to enhance the localization accuracy of the model in object detection. According to the characteristics of *Litchi* small target diseases and pests, the network structure was redesigned to improve the detection effect of small targets. FCOS-FL has a detection accuracy of 91.3% (intersection over union (*IoU*) = 0.5) in the images of five types of *Litchi* leaf diseases and pests, a detection rate of 62.0/ms, and a model parameter size of 17.65 M. Among them, the detection accuracy of *Mayetiola* sp. and *Litchi* algal spot, which are difficult to detect, reached 93.2% and 92%, respectively. The FCOS-FL model can rapidly and accurately detect five common diseases and pests in *Litchi* leaf. The research outcome is suitable for deployment on embedded devices with limited resources such as mobile terminals, and can contribute to achieving real-time and precise identification of *Litchi* leaf diseases and pests, providing technical support for *Litchi* leaf diseases' and pests' prevention and control.

**Keywords:** diseases and insect pests of *Litchi*; FCOS-FL; convolutional block; attention module; G-GhostNet-3.2

## 1. Introduction

*Litchi* (*Litchi chinensis* Sonn) is an evergreen tree belonging to the Sapindaceae family. China is the country with the widest cultivation area and the largest yield of *Litchi* in the world. *Litchi* is rich in nutrients and has a delicious taste. As a cash crop tree species, it is widely planted in South China. In China's mainland, the total output value of *Litchi* was 33.389 billion yuan in 2022 according to the acquisition price [1]. Therefore, promoting the development of the *Litchi* planting industry is beneficial to boosting the economic benefits

of farmers and meeting the market's demands for scaled and high-quality *Litchis* [2,3]. The climate in southern China is usually hot and humid, making *Litchi* susceptible to various pest and disease problems. *Litchi* leaf disease and pest problems can result in a decrease in *Litchi* production and a decline in fruit quality. These issues significantly obstruct the steady development of the *Litchi* industry [4]. The key to solving this problem is to detect, identify, and provide feedback on the disease and insect pests at the early stages of their occurrence. Therefore, timely and efficient detection of *Litchi* leaf diseases and insect pests is an important measure to ensure the yield of *Litchi*. Traditional artificial recognition is greatly influenced by subjective factors and may not be able to successfully diagnose specific diseases, leading to erroneous conclusions and treatment [5,6]. With the rapid popularization of computer technology, it is possible to quickly identify and detect pests and diseases, greatly promoting the development of intelligent agriculture [7].

In early agricultural pest detection research, traditional image processing methods using support vector machine (SVM) [8], K-nearest neighbor (KNN) [9], and other algorithms to detect target pests [10–12] were used for agricultural pest detection. Liu et al. [13] used the histogram of oriented gradients (HOG) feature extraction method to extract features from wheat aphid images in the wheat field, and then used SVM classifiers to identify and locate wheat aphids. Sabro et al. [14] used the Otsu threshold method to segment tomato leaf diseases and insect pests, manually extracting the shape, color, texture, and other characteristics of the disease, and then using a decision tree to classify tomato leaf diseases and insect pests. Pydipati et al. [15] used the color co-occurrence method (CCM) to determine whether texture-based hue, saturation, and intensity (HSI) color features in conjunction with statistical classification algorithms could be used to identify diseased and normal citrus leaves under laboratory conditions. Traditional pest detection methods have achieved some results, but there are still problems such as low efficiency, poor segmentation results, and low recognition rates in complex backgrounds. Therefore, they cannot meet the need for the real-time and accurate detection of *Litchi* leaf diseases and pests in complex field environments [16].

In recent years, significant progress has been made in image recognition technology based on convolutional neural networks [17,18]. Target detection is a branch of image recognition based on convolutional neural network algorithms. A large number of researchers have applied target detection technology to research crop disease and pest detection, making significant breakthroughs [19–21]. Zhang et al. [22] proposed an improved YOLOv5 network that combines DenseNet, attention mechanism, and Bi-FPN to accurately detect unopened cotton bolls in the field at a lower cost. Liu et al. [23] proposed an improved convolutional neural network and a PestNet algorithm with a modular channel attention mechanism to identify 16 pests. Experiments showed that the average accuracy rate reached 75.46%. Dai et al. [24] proposed an improved YOLOv5m method, introducing the Swin Transformer mechanism into the YOLOv5m network to capture more global features and increase receptive fields, enabling more accurate detection of different pests from the dataset. Zhang et al. [25] used the attention method of dynamic mechanism fusion to improve DenseNet and proposed the Dense Channel and Position Self-Attention Fusion Network model, with a high recognition accuracy of 96.90% for six types of navel orange diseases and pests. Liu et al. [26] developed a tomato pest recognition algorithm based on the improved YOLOv4 fusion triple attention mechanism (YOLOv4-TAM), which solves the problem of uneven positive and negative sample numbers in the recognition system and improves the accuracy in identifying tomato pests and diseases.

Research on the intelligent recognition of disease and insect images based on convolutional neural networks has made steady progress in recent years. However, due to the wide variety of diseases and insect pests in *Litchi* and the complex image background, two types of diseases and insect pests, *Mayetiola* sp. and *Litchi* algal spot, are characterized by small and dense pest features, which are likely to cause false detection and missed detection of disease and insect targets, making detection difficult [27,28]. The existing neural network algorithms have high computational complexity, large parameter quantities,

and low detection accuracy for small target diseases and insect pests, such as *Mayetiola* sp. and *Litchi* algal spot. There is still room for improvement in real-time detection and small target disease and insect pest identification. To address the issues of low accuracy and large model parameters in existing methods for detecting *Litchi* leaf diseases and pests, this paper proposes an FCOS-FL model for detecting *Litchi* leaf diseases and pests based on the images of five types of *Litchi* leaf diseases and pests collected from orchards. The proposed model achieves efficient and accurate recognition of the five common types of *Litchi* leaf diseases and pests, with a focus on solving the low detection accuracy of *Mayetiola* sp. and *Litchi* algal spot. This work provides a reference for the real-time and accurate detection of *Litchi* leaf diseases and pests and offers an effective technical support for the prevention and control of *Litchi* leaf diseases and pests. The innovation points of this study include:

(1) Achieving a lightweight model by replacing the backbone network.
(2) Enhancing the features of lychee pests and diseases by adding attention mechanisms.
(3) Improving the generalization of the model by improving the central sampling and central measurement of the model.
(4) Improving the positioning accuracy of the model by improving the loss function.

## 2. Materials and Methods

### 2.1. Dataset Production

#### 2.1.1. Data Collection

The experimental data in this study were collected from three different orchards: the Guangdong Agricultural Technology Extension Station, Guangzhou City, Guangdong Province, China, where the variety collected was glutinous rice dumpling *Litchi*; the State Key Laboratory of South China Agricultural University, Guangzhou City, Guangdong Province, China, where the varieties collected were mainly Gui Wei *Litchi* and glutinous rice dumpling *Litchi*; and the Rijin Orchard in Matouling Farm, Gaozhou City, Guangdong Province and Xiaoliang Town, Maoming City, Guangdong Province, China, where the main variety collected was Gui Wei *Litchi*. The data collection period was from 11 July 2021 to 19 November 2021. Smartphones were chosen as the data collection devices because they meet the convenience requirements of data collection in actual orchard scenarios and can meet the demand for image pixel quality for disease and pest detection. The devices used were iPhone 12 and Xiaomi 6 smartphones. In order to obtain clear images of pests and diseases, the distance between the device and the diseased leaf was set to between 0.2 m and 0.5 m during shooting, and the resolution included $4032 \times 3016$ and $3016 \times 3016$, among others. To more realistically recreate the complex environment of the field and improve the model's generalization, environmental factors were taken into consideration during data collection, including sunny, cloudy, and rainy days; the data collection time period was from 8:30 a.m. to 12:00 p.m. and from 2:00 p.m. to 6:00 p.m. to consider the effect of lighting conditions. To simulate the actual fruit collection process, the shooting angle was set as vertical 90°, upward 45°, and downward 45°. During the collection process, five common characteristics of *Litchi* leaf diseases and pests were identified by visual inspection and by consulting local agricultural experts and farmers [29,30], including *Litchi* sooty mold, *Litchi* anthracnose, *Litchi* algal spot, *Litchi* leaf mite, and *Mayetiola* sp. The characteristics of the five pests and diseases are shown in Table 1. In total, 3900 original images were collected.

Among them, *Litchi* leaf mite has 795 images, and *Litchi* sooty mold has 770 images. *Litchi* anthracnose has 783 images, *Mayetiola* sp. has 738 images, and *Litchi* algal spot has 814 images.

**Table 1.** Characteristics of 5 pests and diseases.

| Name | Characteristics | Picture |
|---|---|---|
| *Litchi* leaf mite | At the initial stage of damage to the leaves, small chartreuse spots appear, and then gradually expand to form irregular large patches. The back of the diseased leaves is fluffy, like felt, uneven, yellow at first, then turns to dark brown to reddish brown, and the back of the leaf margin curls. |  |
| *Litchi* sooty mold | A dark brown to dark brown mold layer forms on the surface of the leaves, which thickens into a coal-smoke-like layer. Scattered black small particles on the mold layer |  |
| *Litchi* anthracnose | Brown spots are produced at the leaf tips and edges, and the brown spots on the leaves are circular in shape, followed by the formation of small black particles. |  |
| *Mayetiola* sp. | By using larvae to grow hoops, the affected spots gradually cause protrusions on the surface of the leaves, forming nodules. In severe cases, there are more than ten or even hundreds of galls on a single leaf, causing the leaf to twist. As the leaves age and dry up, some of them become perforated. |  |
| *Litchi* algal spot | When the disease occurs, the leaves have black brown or brick red spots, and there are gray–green or yellow–brown plush-like substances on the disease spots. Spindle-shaped to short-strip-shaped black spots often appear on both sides of the midvein of the leaf, with a gray white center |  |

### 2.1.2. Data Cleaning and Labeling

In order to reduce the number of repetitive and blurred images, the method of manual selection was used to clean the collected images, i.e., images that were blurry due to handheld shooting or images with unclear features of *Litchi* leaf diseases and pests were deleted. The final dataset consisted of 3725 images, including 1061 images taken in direct

sunlight, 960 images taken against backlight, 720 images taken after rain, and 984 images taken on cloudy days.

This experiment focuses on visible light images of *Litchi* leaf diseases and pests and aims to design a recognition model for *Litchi* leaf diseases and pests. The collected *Litchi* images require manual annotation to mark the regions with *Litchi* leaf diseases and pests, and to write the coordinates of the marked regions into XML files. Then, these annotated images can be fed into the neural network model to learn the features of *Litchi* leaf diseases and pests. In this study, the open-source annotation software "labelImg" was used for manual annotation of the *Litchi* leaf disease and pest dataset. LabelImg was developed by Chinese computer scientist Yude Wang, and the version used in this article is V1.8.5. With this software, the target region of *Litchi* leaf diseases and pests can be selected and labeled with its type of disease or pest, and an XML file containing the location coordinates and disease or pest categories of the target box can be obtained for neural network model training. The annotation process is illustrated in Figure 1.

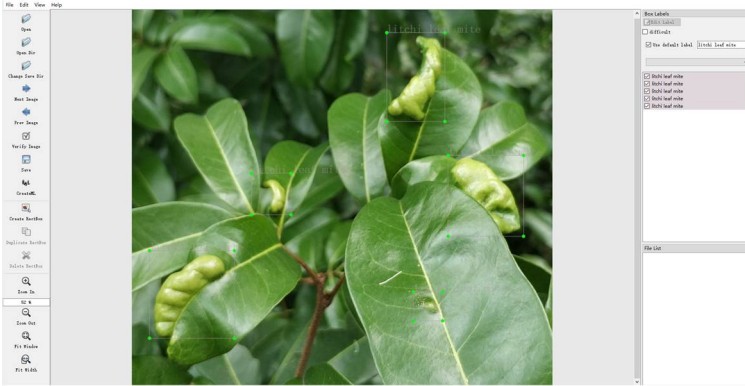

**Figure 1.** LabelImg annotation software interface diagram.

2.1.3. Dataset Partitioning

After annotating the *Litchi* diseases and pests dataset using the labeling software LabelImg, the images and label files were divided into training and testing sets according to the requirements of different object detection models. The *Litchi* disease and pest image data were randomly divided into training sets, validation sets, and test sets at a ratio of 8:1:1. The basic information of the dataset is shown in Table 2.

**Table 2.** Dataset of *Litchi* diseases and pests.

| Category | Training | Validation | Test | Total |
|---|---|---|---|---|
| *Litchi* leaf mite | 630 | 79 | 79 | 788 |
| *Litchi* sooty mold | 577 | 72 | 72 | 721 |
| *Litchi* anthracnose | 567 | 71 | 71 | 709 |
| *Mayetiola* sp. | 636 | 80 | 80 | 796 |
| *Litchi* algal spot | 569 | 71 | 71 | 711 |
| Total | 2979 | 373 | 373 | 3725 |

*2.2. Experimental Protocol*

This paper proposes a *Litchi* leaf diseases and pests detection method that consists of three main steps: data collection and processing, model construction and optimization, and detection result output, as illustrated in Figure 2. Firstly, a *Litchi* leaf diseases and pests dataset is constructed by collecting and cleaning image data from field orchards, followed by manual annotation. Then, a *Litchi* leaf diseases and pests recognition model based on the FCOS model is designed to address the challenges of recognizing small targets with multiple categories and improving the model's lightweight performance for practical application. By improving the FCOS model, a *Litchi* leaf diseases and pests detection model

based on FCOS is constructed to achieve accurate and efficient identification of diseases and pests on *Litchi* leaves, providing technical support for the prevention and control of *Litchi* diseases and pests and improving the yield and quality of *Litchi*.

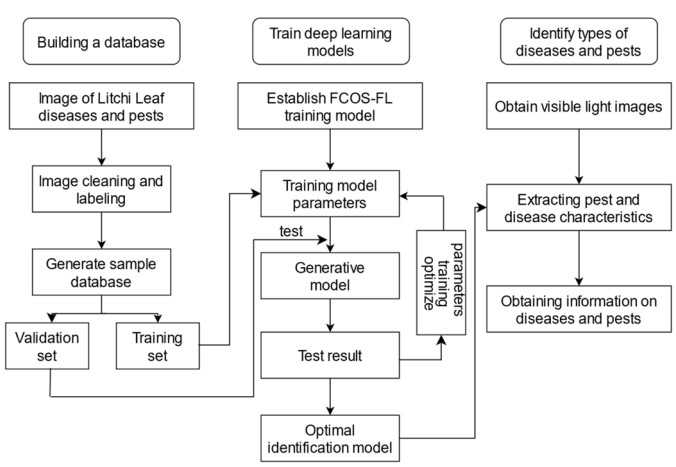

**Figure 2.** The experimental protocol workflow diagram.

*2.3. FCOS-FL Construction*

2.3.1. FCOS Model

The FCOS algorithm is a fully convolutional, anchor-free, single-stage target detection network proposed by Tian et al. [31]. It is one of the representative methods for target detection based on deep learning, and it has achieved good results in conventional target detection tasks. The basic idea of the FCOS algorithm is to achieve the goal of target detection by classifying and regressing each position in the image. The FCOS network structure is shown in Figure 3. The FCOS network mainly includes three parts: the feature extraction backbone of the residual neural network (ResNet) [32]; the feature pyramid network (FPN) for object detection [33]; FCOS for header detection.

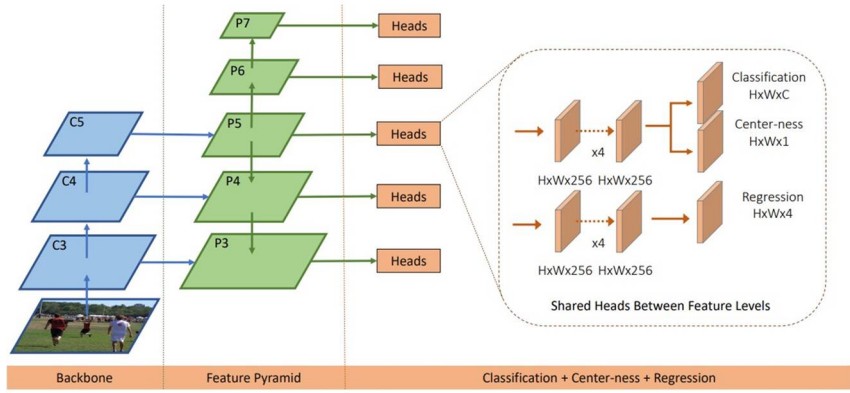

**Figure 3.** The FCOS model.

Specifically, the *Litchi* leaf diseases and pests dataset serves as the input, and a series of convolutional and pooling operations are performed to extract the features of the manually annotated target areas of the diseases and pests. These feature maps are then passed into a residual network for further feature extraction, in order to accurately identify the diseases and pests. Next, the feature maps are passed into a feature pyramid network, which upsamples the maps to different resolutions to adapt to the uneven distribution of sizes in the *Litchi* leaf diseases and pests. Finally, each feature map is classified, regressed, and center point detected to determine whether a target is present at that location. The regressor predicts the target box and detects the center point of each disease or pest target

to determine the precise location. The model assigns a confidence score to each location and category to indicate whether it belongs to a certain type of disease or pest. Based on the score, the most likely target is determined and information such as its category, location, and confidence score is output.

### 2.3.2. G-GhostNet

HAN [34] proposed the improved G-GhostNet in 2022. This article uses G-GhostNet-3.2 as the backbone network extraction feature for improving FCOS. G-GhostNet is a new lightweight network architecture optimized for server GPUs. G-GhostNet utilizes neural networks to extract effective redundant feature maps in each stage, with the feature map size unchanged between different structural blocks. The structure is shown in Figure 4. The redundant feature map obtained directly from shallow structural blocks in Figure 3a lacks high semantic information; thus, as shown in Figure 4b, a mix operation is added for information compensation. The mix operation with information compensation is depicted in Figure 4c.

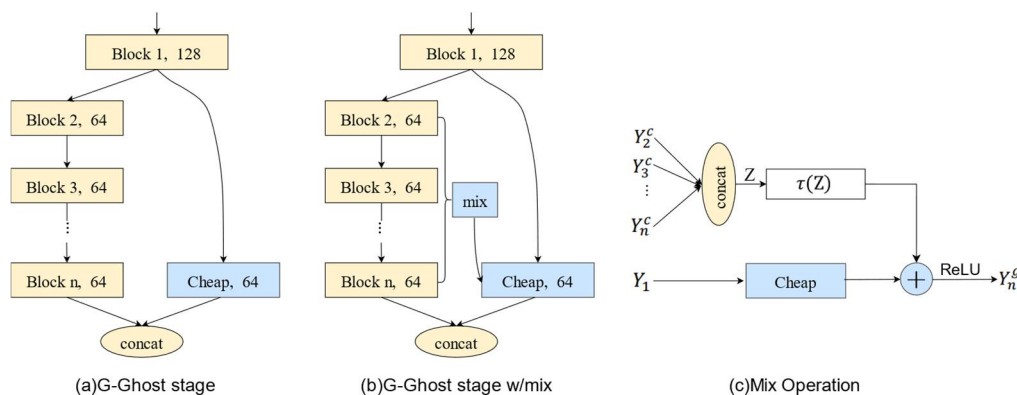

**Figure 4.** G-GhostNet low-cost operation.

The mix operation of information compensation is shown in Figure 4c, where $Y_i^c$ represents the output characteristic diagram of the i-th structural block, and the stacked characteristic diagram is represented as $Z \in R^{c' \times h \times w}$. This is followed by $\tau(Z)$ average pooling and convolution, which can be expressed as follows:

$$\tau(Z) = W \times \text{Pooling}(Z) + b. \tag{1}$$

From the perspective of practical application of the model, this paper considers the difficulty of deploying the model on mobile and embedded devices. To address this issue, the G-GhostNet module, which is characterized by its versatility and scalability, is introduced into the FCOS model to reduce the computational and parameter complexity of the FCOS-FL model and optimize its lightweight design.

### 2.3.3. Attention Mechanism

The attention mechanism is a method that utilizes global pooling operations to extract global information from a feature, and then applies the global information to the feature graph. This method utilizes global maximum pooling and average pooling to extract global information, enhancing the ability of network feature extraction. Wu Tong et al. [35] proposed a central moment pooling spatial attention mechanism based on statistical analysis.

This paper addresses the challenge of recognizing *Litchi* leaf diseases and pests due to their small size and high difficulty. To enhance the recognition accuracy of disease and pest targets, this paper combines spatial attention mechanism with central moment pooling, which provides more statistical information on the features of *Litchi* leaf diseases and pests. Specifically, the central moment pooling attention (CMPA) mechanism is introduced into the FCOS model to enhance the *Litchi* leaf disease and pest features using a superior spatial

attention mechanism. The structure of CMPA is shown in Figure 5. By adopting this approach, the model achieves better recognition accuracy for *Litchi* leaf diseases and pests.

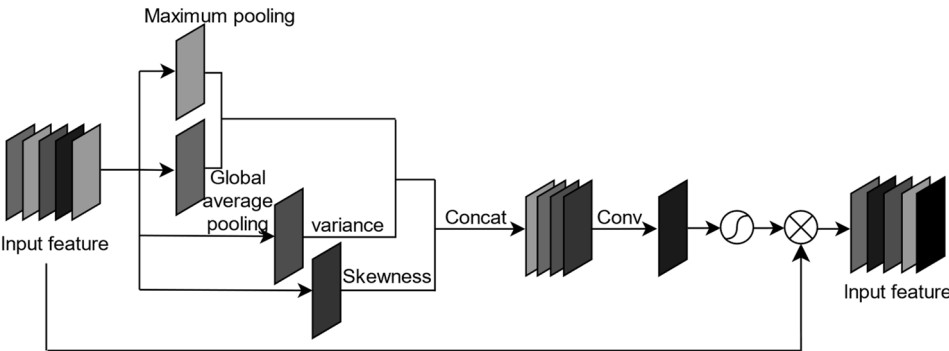

**Figure 5.** The CMPA model.

2.3.4. Improved Center Sampling and Center Metrics

In response to the uneven size and significant differences in the morphology of *Litchi* leaf diseases and pests, using the original center sampling formula can lead to inaccurate recognition of the target. As shown in Figure 6b, if the original center sampling method is used, sampling is conducted using a fixed square, which may miss some positive sample points [36] for targets with significant differences in width and height, resulting in a positive sample imbalance. To address this issue, an improved center sampling method that can adaptively adjust based on the aspect ratio of the target is proposed, as shown in Figure 6c. This method increases the number of positive samples during the training process and improves the problem of positive sample imbalance.

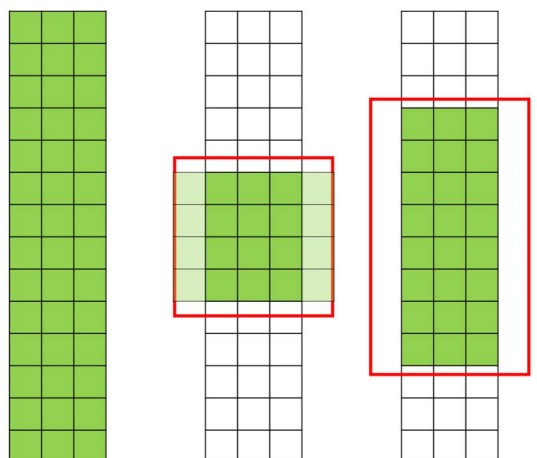

(a) Initial sampling　　(b) Central sampling　(c) Improved center sampling

**Figure 6.** The sampling methods of FCOS.

This paper proposes a new center measurement method to address the problem of predicting bounding boxes for *Litchi* leaf disease and pest targets with large differences in width and height. The new centerness formula is designed as Equation (2). In the Equation (2), $w$ and $h$ represent the width and height of the true bounding box of the object, respectively. $l^*$, $t^*$, $r^*$, and $b^*$ respectively denote the distances between a certain positive sample point and the left, top, right, and bottom edges of the true bounding box.

$$Centerness^* = \left( \frac{\min(l^*, r^*)}{\max(l^*, r^*)} \times \frac{\min(t^*, b^*)}{\max(t^*, b^*)} \right)^{\frac{1}{2} \times \frac{\min(w,h)}{\max(w,h)}} \tag{2}$$

This paper addresses the problem of box regression for *Litchi* leaf disease and pest targets with large aspect ratios. The original centerness metric is not sensitive enough to the aspect ratio of the targets, and when a sample point is located far from the center of the longer side but still within the center of the shorter side (as shown in the green grid of Figure 7a, its centerness value is lower, and its predicted box score is forced to decrease, leading to excessive suppression of the sample point [37]. As shown in (b) of Figure 7, based on Equation (2), when there are *Litchi* leaf diseases and pests targets with large aspect ratios in the image, the centerness metric of the above sample point will not be too low, and its predicted box will not be excessively suppressed. The improved centerness can still suppress the effect of sample points located at the edge, making FCOS pay more attention to the center of the *Litchi* leaf diseases and pests targets, thereby improving the accuracy of box regression.

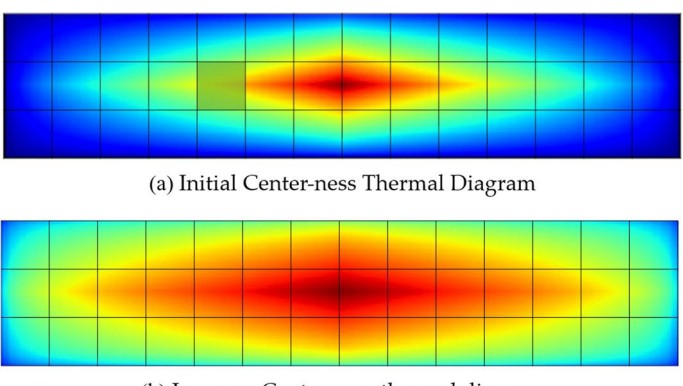

(a) Initial Center-ness Thermal Diagram

(b) Improve Center-ness thermal diagram

**Figure 7.** Improved centerness thermal diagram before and after.

2.3.5. Improved Loss Function

The loss function is a function used to measure the difference between the model's predicted results and the true results, and can guide the model's optimization and evaluate its performance. The original model used the *IoU* loss to perform regression training, as shown in Equation (3).

$$L_{IoU} = -\ln(IoU). \tag{3}$$

$$L_{reg} = L^*_{IoU} = (\alpha - IoU^\gamma)L_{IoU}. \tag{4}$$

The original *IoU* loss did not reflect the true differences between the width, height, and confidence of the predicted bounding boxes, leading to errors in the predicted boxes. To address the problem of *Litchi* leaf diseases and pests being densely connected and difficult to distinguish, resulting in significant differences between positive and negative samples in small target images, this paper introduces the concept of hard example mining to focus the training process on beneficial samples, thereby improving the accuracy of the model in recognizing and locating difficult to identify disease and pest targets. The improved loss function is expressed as Equation (4).

Similarly to focal loss, $(\alpha - IoU^\gamma)$, it is mainly used for difficultly divided sample mining, which is herein referred to as difficultly divided sample weight. A larger *IoU* value of a sample and a real target indicates more accurate positioning. At this point, the sample is defined as a separable sample. When the *IoU* value is smaller, it indicates that its positioning is more inaccurate. At this time, the sample is defined as hard to distinguish. In order for the model to focus on the location and regression of difficultly divided samples during the training process, an adaptive weight is added to the original loss, and the model detection accuracy is improved by amplifying the location loss of difficultly divided samples.

### 2.3.6. Detection Model of *Litchi* Leaf Diseases and Pests Based on FCOS-FL

The FCOS-FL *Litchi* leaf disease and pest detection model proposed in this article is shown in Figure 8. The original ResNet-50 is replaced with G-GhostNet-3.2 as the backbone network to extract features, reducing the weight of FCOS and accelerating detection speed. A central moment pooling attention mechanism (CMPA) is added to the last three stages of G-GhostNet-3.2 to enhance the feature maps. Using the width and height information of real targets to improve the center sampling and center measurement of FCOS, it is possible to obtain more positive samples based on the width and height of real targets during model sampling, thus improving the center measurement, avoiding excessive suppression of positive samples of targets with large width and height differences, and improving the generalization of FCOS. An improved location loss function is proposed, which adds difficultly divided sample weights to the FCOS training process to improve the location accuracy of the model. Aiming at the small size of *Litchi* leaf diseases and insect pests, the FCOS detection layer was adjusted, and the outputs from C2 to C5 in the last four stages of the backbone network G-GhostNet-3.2 were extracted as the input characteristic map of the FPN. The two downsamples in the FPN were changed to one downsample to obtain P6, and P3 was upsampled and fused with C2 to obtain P2. The input of the detection head was adjusted to P2 to P6, maintaining five layers.

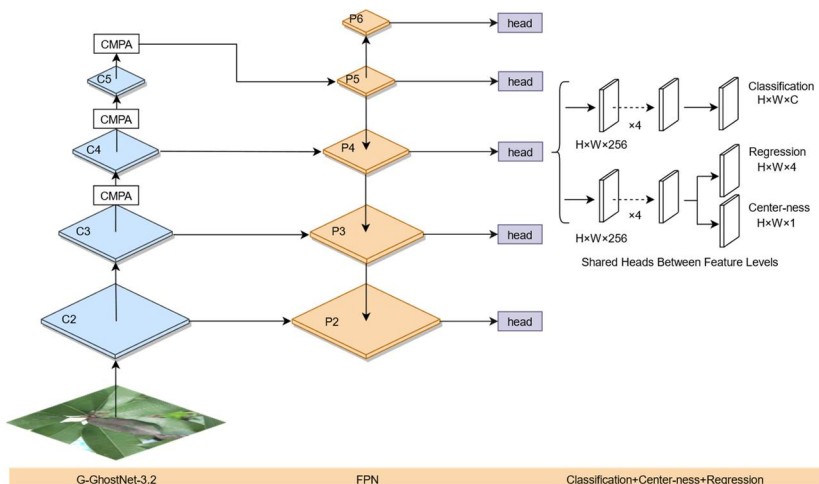

**Figure 8.** The FCOS-FL model.

### 2.4. Training Environment and Parameter Settings

The server hardware configuration and software environment configuration used in the experiment are shown in the Table 3.

**Table 3.** The configuration of server hardware and software environment.

| Hardware Configuration | Software Environment |
|---|---|
| CPU: Inter(R) Xeon(R) Gold 6240 CPU@2.59 GHz | OS: Windows10 |
| Memory: 192 GB | Pycharm2021.3.3 + python 3.8.10 |
| Disk: 4 TB | CUDA 11.1 + CuDNN 8.0.5 |
| GPU*2: NVIDIA GeForce GTX 3090Ti | PyTorch 1.8.0 |

This article used an enhanced training set of 3725 *Litchi* images for model training. To enhance test reliability, the image resolution was uniformly scaled to $512 \times 512$ before inputting network model training. A single graphics card was used for training. The training involved 200 rounds, the batch size was set to 8, the number of iterations was set to 48, and the initial learning rate was set to 0.01. Warmup [38] was used to warm up the learning rate during training to improve the stability of the model. The cosine

annealing strategy was used to update the learning rate. The optimizer used SGD [39] to adjust the error. When evaluating the detection performance of the trained model, the detection results were postprocessed using NMS with non-maximum suppression, and the *IoU* threshold of NMS was set to 0.5.

*2.5. Model Evaluators*

In this study, precision, recall, average precision(AP), mean average precision (mAP), parameters (Params), and average time per single image in the test set (Time) were used as model evaluation metrics. Recall and precision are two of the important indexes for evaluating the model. The larger the area of the precision–recall curve, the better the comprehensive performance of the model. mAP is a measure of detection accuracy in target detection, and the higher the mAP, the better the detection effect of the model. Model parameter quantity (Params) is the quantity of parameters contained in the model structure, and the smaller the parameter quantity, the smaller the memory space required for model operations.

## 3. Experiments and Results

*3.1. Ablation Contrast Experiment*

In order to verify the effectiveness of the various improvement strategies proposed, a series of ablation experiments were conducted in this paper. The network model was trained using the same training parameters on the same platform and under the same experimental framework. This mainly included a comparison of different backbone networks, the exploration of the optimal location of the central moment pooling attention mechanism (CMPA) in the backbone network, the exploration of parameter settings for difficult sample weights in the improved loss function, the comparison of various improvement schemes in FCOS and FCOS-FL, and the comparison of different models.

3.1.1. Comparison of Different Backbone Networks

In order to verify the effectiveness of using lightweight convolutional neural network G-GhostNet-3.2 as the backbone network of FCOS, different backbone networks were introduced in FCOS models, and then trained on the constructed *Litchi* leaf disease and pest dataset. The comparison results of the detection performance of FCOS under different backbone networks are shown in Table 4.

**Table 4.** The detection results of different backbone networks.

| Backbone Network | Precision (%) | | | Time (ms) | Params (M) |
|---|---|---|---|---|---|
| | AP | AP$_{50}$ | AP$_{75}$ | | |
| ResNet-50 | 59.4 | 87.1 | 64.1 | 76.6 | 32.12 |
| EfficientNet-b2 | 53 | 81.5 | 54.1 | 62.1 | 15.03 |
| GhostNet | 52.1 | 82.9 | 55 | 49.7 | 10.3 |
| MobileNet-v2 | 58.7 | 87.1 | 63.6 | 51.5 | 10.04 |
| G-GhostNet-3.2 | 60.2 | 88.4 | 66.8 | 53.7 | 17.62 |

Note: Precision means the proportion of correct predictions in the positive samples predicted by the classifier. Rate means average time per single image in the test set. Params means the number of model parameters.

As can be seen from Table 4, when the *IoU* threshold was 0.5 and 0.75, and the average of other *IoU* thresholds was AP, G-GhostNet-3.2 had the highest detection accuracy compared to other backbone networks, reaching 88.4%, 66.8%, and 60.2%, respectively. Among them, the accuracy of the original FCOS using ResNet-50 as the backbone network was second only to G-GhostNet-3.2, but its disadvantage was that the number of parameters, amount of computation, and detection time were higher than those of other backbone networks. In addition, the parameter number and calculation amount of GhostNet were not the lowest in this comparison, but the detection time was the lowest, while the detection accuracy was not inferior to EfficientNet-b2, proving its efficiency. Although G-GhostNet-

3.2 was not superior to MobileNet-v2 in terms of its light weight, it achieved a greater degree of light weight based on improved accuracy compared to ResNet-50 of the initial FCOS. Figure 9 shows the performance of the G-GhostNet backbone network in terms of loss and $AP_{50}$. It can be seen from the figure that the G-GhostNet network backbone performed best. The above comparative experiments fully prove that using G-GhostNet-3.2 as the backbone network to extract features could meet the real-time requirements of *Litchi* leaf disease and pest detection tasks, with a greater advantage in detection accuracy, which is significant for improving FCOS.

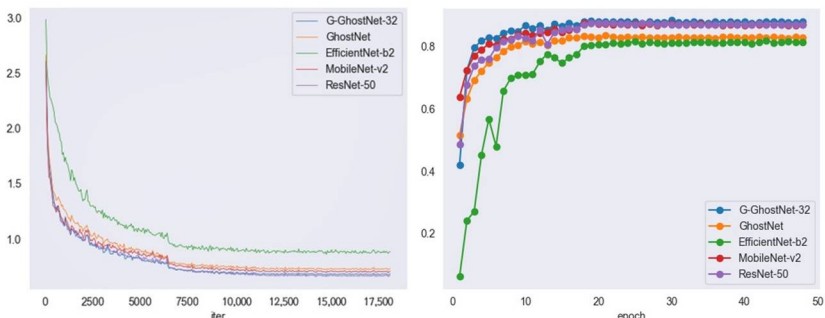

**Figure 9.** Loss (**left**) and $AP_{50}$ (**right**) variation in different backbone networks.

### 3.1.2. Comparison of CMPA at Different Locations

To verify the effectiveness of the central moment pooling attention mechanism (CMPA) and to explore its effectiveness in feature enhancement in FCOS models, comparative experiments were conducted with CMPA placed in different locations. The results are shown in Table 5.

**Table 5.** The detection results of CMPA at different locations.

| Stage | Recall (%) | $AP_{50}$ (%) | $AP_{50}$ (%) | | | | |
|---|---|---|---|---|---|---|---|
| | | | *Litchi* Leaf Mite | *Litchi* Sooty Mold | *Litchi* Anthracnose | *Mayetiola* sp. | *Litchi* Algal Spot |
| C5 | 93.9 | 88.3 | 83.9 | 95.2 | 92.1 | 87.1 | 83.2 |
| C3 + C4 | 94.1 | 88.0 | 84.3 | 95.3 | 91.9 | 86.2 | 82.3 |
| C3 + C5 | 93.8 | 88.8 | 84.7 | 95.3 | 91.6 | 88.4 | 83.8 |
| C4 + C5 | 93.3 | 88.6 | 84.0 | 95.9 | 91.7 | 87.1 | 84.2 |
| C3 + C4 + C5 | 93.9 | 89.3 | 86.3 | 94.9 | 91.9 | 87.6 | 85.8 |
| C2 + C3 + C4 + C5 | 93.2 | 88.7 | 84.7 | 95.5 | 91.5 | 87.4 | 84.2 |

Note: Recall means evaluate the percentage of coverage predicted for all actual positive examples. $AP_{50}$ means the detection accuracy of the model at an *IoU* threshold of 0.5.

As can be seen from Table 5, when the CMPA was placed in the last three stages, the average $AP_{50}$ of various pests and diseases reached the highest level at 89.3%. Moreover, the detection effect of difficult-to-detect *Mayetiola* sp. and *Litchi* algal spot pest and disease samples was excellent. Among them, *Litchi* leaf mite and *Litchi* algal spot had the highest detection effect on $AP_{50}$, with 86.3% and 85.8%, respectively. *Litchi* anthracnose and *Mayetiola* sp. were second only to the highest detection rates of 92.1% and 88.4%, with detection results of 91.9% and 87.6%. Although the detection accuracy of *Litchi* sooty mold differed by 1% from the optimal accuracy, this type of leaf disease and insect pest is relatively easy to separate from the other four types, thus having little impact on the overall detection effect. Therefore, this article proposes adding the CMPA to the last three stages of the backbone network G-GhostNet-3.2 to provide a better input characteristic graph for FPN.

### 3.1.3. Comparison of Improved Loss Functions with Different Hyperparameters

In order to verify the effectiveness of the improved loss function and explore the optimal combination of different hyperparameters that are difficult to distinguish between sample weights in the improved loss function, the comparison results are shown in Table 4. First, the weight of difficultly divided samples $(\alpha - IoU^{\gamma})$ was fixed. A $\gamma$ value of 1 indicates that the weight of difficultly divided samples only varied linearly, without influencing the size of the *IoU*. Thus, a smaller value of 1.5 was considered to compare the differences in $\alpha$. Then, the value of $\alpha$ was fixed to compare differences in $\gamma$. During the test, when the detection performance of the target detection model FCOS improved to a certain extent and started to decline, the test was stopped. A higher value of $\gamma$ was achieved when $\alpha$ increased to 4, after which there was a downward trend in detection performance.

From Table 6, it can be seen that, when $\alpha = 3$, AP and $AP_{75}$ reached the highest level, whereas $AP_{50}$ and recall began to show a downward trend. When $\alpha = 4$, all indicators began to decline. When $\alpha = 2$, although $AP_{50}$ and recall reached the highest level, overall superior results were obtained when $\alpha = 3$, at which point the maximum improvement in FCOS could be achieved with $AP_{50}$ reaching a maximum of 89.4%. Therefore, in this paper, the hard-to-distinguish sample weight hyperparameters of the loss function $\alpha$ and $\gamma$ were set to 3.

**Table 6.** The parameter results for the weight of difficult samples.

| Alpha | Gamma | AP (%) | $AP_{50}$ (%) | $AP_{75}$ (%) | Recall (%) |
|-------|-------|--------|--------------|--------------|------------|
| 1.5 |     | 59.5 | 88.6 | 66.0 | 93.1 |
| 2 |       | 60.6 | 88.9 | 67.4 | 93.8 |
| 3 | 1.5   | 62.2 | 88.4 | 69.5 | 93.5 |
| 4 |       | 62.2 | 88.4 | 69.4 | 94.0 |
|   | 2     | 61.2 | 88.3 | 67.5 | 93.3 |
| 3 | 3     | 61.2 | 89.4 | 68.1 | 94.1 |
|   | 4     | 60.9 | 88.6 | 67.3 | 94.1 |

Note: AP (average precision) means the area under the P–R curve for a single category. This index can comprehensively indicate the precision and recall of a model. $AP_{75}$ means the detection accuracy of the model at an *IoU* threshold of 0.75.

### 3.1.4. Comparison of Different Improvement Methods

The effectiveness of the improved algorithm's introduction of the CMPA attention mechanism module and the loss function optimization method was verified by conducting ablation experiments on the improved FCOS algorithm. The test results are shown in Table 7.

**Table 7.** The results of different improvement methods.

| Backbone Network | A | B | C | D | AP (%) | $AP_{50}$ (%) | $AP_{75}$ (%) | Recall (%) |
|------------------|---|---|---|---|--------|--------------|--------------|------------|
| G-GhostNet-3.2 |   |   |   |   | 60.2 | 88.4 | 66.8 | 92.4 |
| G-GhostNet-3.2 | √ |   |   |   | 60.9 | 89.3 | 66.9 | 93.9 |
| G-GhostNet-3.2 |   | √ |   |   | 60.2 | 89.0 | 67.6 | 94.1 |
| G-GhostNet-3.2 |   |   | √ |   | 61.2 | 89.4 | 68.1 | 94.1 |
| G-GhostNet-3.2 |   |   |   | √ | 63.0 | 90.7 | 72.7 | 95.0 |
| G-GhostNet-3.2 | √ | √ | √ | √ | 65.1 | 91.3 | 73.2 | 96.2 |

A: CMPA was added to the last three stages of G-GhostNet-3.2; B: improved center sampling and center measurement with width and height adaptability; C: modified loss function with the weight of difficult samples; D: adjusted detection layer of FCOS.

As can be seen from Table 7, after adding the four improvements, the model $AP_{50}$ increased to 91.3%. Specifically, the accuracy rate of the FCOS model that only replaced the backbone network with G-GhostNet-3.2 increased by 0.9 percentage points. After adding

the four improvements, the accuracy rate of the model reached 91.3%, i.e., 2.9 percentage points higher than the original model, verifying the effectiveness of the improved method.

### 3.2. Comparison of Detection Effects between FCOS and FCOS-FL

The experimental data were collected in four weather conditions: direct sunlight, backlight, after rain, and overcast days. The collected image data were experimentally compared using the original FCOS and FCOS-FL, and the results are shown in Figure 10. In Figure 10, the yellow box is marked as missed inspection. The red box in Figure 10a shows *Litchi* leaf mite. The purple block box in Figure 10b shows *Litchi* algal spot, the light blue box in Figure 10c shows Mayetiola sp, and the dark blue box in Figure 10d shows *Litchi* anthracnose. As can be seen from Figure 10, the original FCOS model had different degrees of missed detection when detecting various pests and diseases. When exposed to direct sunlight, *Litchi* leaf mites could cause FCOS to miss detection due to interference from light spots, whereas FCOS-FS could eliminate the interference from light spots for accurate detection (Figure 10a). For *Mayetiola* sp. and *Litchi* algal spot (Figure 10b,c), FCOS experienced a large number of missed detections due to their extremely small and dense disease volume, whereas FCOS-FL performed excellently in detecting these two small target pests and diseases. In addition, FCOS-FL could accurately identify multiple pests and diseases on a single leaf (Figure 10d).

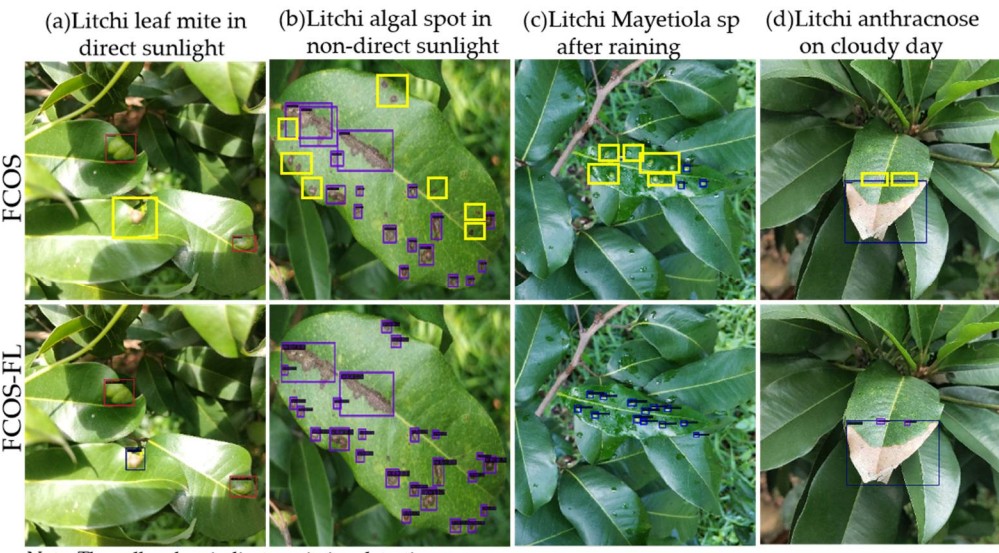

Note: The yellow box indicates missing detection

**Figure 10.** Detection results of FCOS and FCOS-Fl under different weather conditions.

### 3.3. Performance Comparison of Different Models

To verify the effectiveness of the FCOS-FL model proposed in this article, comparative tests were conducted with faster R-CNN, RetinaNet, VFNet, and the original FCOS model under the same training conditions. The comparison of detection performance is shown in Table 8.

**Table 8.** The detection results of different networks.

| Network Model | Time (ms) | Params (M) | AP$_{50}$ (%) | AP$_{50}$ (%) | | | | |
| --- | --- | --- | --- | --- | --- | --- | --- | --- |
| | | | | *Litchi* Leaf Mite | *Litchi* Sooty Mold | *Litchi* Anthracnose | *Mayetiola* sp. | *Litchi* Algal Spot |
| Faster R-CNN | 86.1 | 41.14 | 75.2 | 82.5 | 94.2 | 90.5 | 47.8 | 56.5 |
| RetinaNet | 80.1 | 36.19 | 77.1 | 83.1 | 93.6 | 88.6 | 51.8 | 68.2 |
| VFNet | 84.3 | 32.49 | 71.7 | 72.9 | 89.2 | 86.4 | 51.0 | 59.0 |
| FCOS | 76.6 | 32.12 | 87.1 | 85.4 | 95.3 | 90.4 | 83.8 | 81.0 |
| FCOS-FL | 62.0 | 17.65 | 91.3 | 84.8 | 95.7 | 90.9 | 93.2 | 92.0 |

As can be seen from Table 8, compared to faster R-CNN, RetinaNet, and VFNet, the accuracy of FCOS-FL in terms of AP$_{50}$ increased by 16.1, 14.2, and 19.6 percentage points, respectively, with respect to model detection accuracy, while it increased by 4.2 percentage points compared to the original FCOS. In terms of detection speed and model parameter size, FCOS-FL improved the detection rate by 19.1% and reduced the model parameter size by 45% compared to the original FCOS. In FCOS-FL, the detection accuracy of four pests and diseases (*Litchi* sooty mold, *Litchi* anthracnose, *Mayetiola* sp., and *Litchi* algal spot) was the best. For *Mayetiola* sp. and *Litchi* algal spot, with high detection difficulty, FCOS-FL improved the detection accuracy by 9.4 and 11 percentage points compared to the original FCOS, respectively, demonstrating the superiority of FCOS-FL in the detection of small target pests and diseases. The above comparison verifies that FCOS-FL achieved a lightweight model while improving detection accuracy.

## 4. Discussion

*Litchi* diseases and pests can reduce yield, affect the quality of *Litchi*, and reduce the benefits of farmers. However, there are many types of *Litchi* diseases and pests, and their characteristics are small, making detection difficult. Currently, the main method for detecting *Litchi* diseases and pests in orchards is still visual identification, but deep learning target detection methods have been widely applied to *Litchi* detection [40–43]. However, research on the application of deep learning target detection methods to *Litchi* diseases' and pests' detection is relatively scarce. In view of these problems, this paper further optimized the model structure based on the advantages of existing research at the network design level, and proposed a strong targeted model optimization method for the characteristics of the five *Litchi* leaf diseases and pests. The FCOS-FL model was experimentally proven to be superior in terms of accuracy and lightweight performance compared to the pre-improved model on the self-built *Litchi* leaf diseases and pests dataset. This method can meet the real-time detection needs of *Litchi* leaf diseases and pests, and the model can be deployed on embedded resource-limited devices such as mobile terminals to achieve real-time recognition of *Litchi* leaf diseases and pests, providing a new solution for the prevention and control of *Litchi* diseases and pests. The overall experimental conclusions are as follows:

(1) Regarding the lack of research on deep learning in *Litchi* diseases and pests and the low detection accuracy of the existing FCOS model in detecting *Litchi* leaf diseases and pests, the FCOS-FL model proposed in this paper adds an attention mechanism and redesigns the network structure to achieve an average detection accuracy of 91.3% on the detection of five types of *Litchi* leaf diseases and pests under different natural environments on the AP$_{50}$, which is 4.2% higher than the detection accuracy of the original FCOS model. Among them, the detection accuracy of the small target pests with high detection difficulty, Mayetiola sp. and *Litchi* algal spot, reached 93.2% and 92.0%, respectively. It proves that the detection accuracy of the proposed FCOS-FL model is sufficient for detecting five types of *Litchi* leaf diseases and pests.

(2) To solve the problem of the large parameter size of the original FCOS model, G-GhostNet-3.2 was used as the backbone network to achieve a model that was lightweight. The parameter size of the FCOS-FL model proposed in this paper is only 17.65 M, which is 45% smaller than the parameter size of the original FCOS model. The single-image detection speed is 62.0 ms, which proves that the proposed model is suitable for deployment on embedded resource-limited devices such as mobile terminals for real-time and fast recognition and detection of *Litchi* diseases and pests.

(3) The proposed FCOS-FL model outperforms other commonly used models, including Faster R-CNN, RetinaNet, VFNet, and the original FCOS model, in terms of detection accuracy and model parameter size. These results show that the FCOS-FL model has higher accuracy and smaller model parameters, providing more reliable support for real-time and accurate detection of the five types of *Litchi* diseases and pests.

(4) In summary, the *Litchi* diseases and pests detection model FCOS-FL proposed in this paper can meet the real-time recognition of *Litchi* diseases and pests in practical applications. The recognition effect and model lightweight degree perform well on the self-built *Litchi* diseases and pests dataset, which can provide a reference for the prevention and control of *Litchi* diseases and pests. Compared with other *Litchi* diseases and pests detection studies, the model and optimization methods used in this study are more in line with the disease characteristics, and the constructed dataset has more complete types of diseases. However, there are still shortcomings in this study. The dataset collected in this study still has room for improvement, and future research should collect more varieties and full-cycle *Litchi* images to supplement the dataset. There are still a small number of missed detections during the detection process, and further optimization of the detection model should be performed to improve its accuracy. Future work should consider deploying the model to edge computing devices for field applications.

**Author Contributions:** Conceptualization, J.X., X.Z. and Z.L.; methodology, J.X., X.Z. and Z.L.; software, X.Z. and Z.L.; validation, X.Z., Z.L. and F.L.; formal analysis, F.L. and Z.L.; investigation, J.X., X.Z. and Z.L.; resources, J.X. and W.W.; data curation, F.L.; writing—original draft preparation, J.X. and X.Z.; writing—review and editing, J.X.; visualization, X.Z. and Z.L.; supervision, W.W. and J.L.; project administration, W.W. and J.L. All authors have read and agreed to the published version of the manuscript.

**Funding:** This research was funded by the Co-constructing Cooperative Project on Agricultural Sci-tech of New Rural Development Research Institute of South China Agricultural University (No. 2021XNYNYKJHZGJ032). It was also partly supported by the China Agriculture Research System of MOF and MARA, China (No. CARS-32-11); the Guangdong Provincial Special Fund for Modern Agriculture Industry Technology Innovation Teams, China (No. 2023KJ108); the Laboratory of Lingnan Modern Agriculture Project, China (No. NT2021009); the Guangdong Province Rural Revitalization Strategy Projects (No. TS-1-4); and the Guangdong Science and Technology Innovation Cultivation Special Fund Project for College Students ("Climbing Program" Special Fund), China (No. pdjh2023a0074 and No. pdjh2021b0077).

**Data Availability Statement:** The data presented in this study are available on request from the corresponding author. The data are not publicly available due to the privacy policy of the organization.

**Acknowledgments:** The authors would like to thank the anonymous reviewers for their critical comments and suggestions for improving the manuscript.

**Conflicts of Interest:** The authors declare no conflict of interest.

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
