# Peer review of "Detection of Litchi Leaf Diseases and Insect Pests Based on Improved FCOS"

_agronomy, doi:10.3390/agronomy13051314_

Round 1

Reviewer 1 Report

The topic is interesting and relevant. However, the quality of the presentation is extremely low.  This is not a computer science or mathematical journal but "agronomy". The presentation of only mathematical definitions and equations instead of presenting the method steps is not appropriate.  After all, these definitions can be found in textbooks or other publications. There is a need to present a research design or at least present the method steps of how the data was collected. Readers want to know how many sites/orchards were involved. Provide details on how the data was processed and how the performance of the methods was judged.

Abstract:

Include a sentence or two in line 15 indicating why Litchi disease was important to investigate. Also, include a sentence in the abstract summarizing how the fieldwork was undertaken.  

Write the meaning of FCOS in full in the abstract

The abstract must have a conclusion or take-home message in line with your aim/objective. 

Introduction:

The introduction is well-written. A few points to improve it:

Include a sentence or two about the economic impacts of Litchi or the diseases under investigation. This will strengthen the motivation for this study. 

In line 50, write HOG in full.

The normal style is to first present the name in full and then followed by its abbreviation in brackets the first time it is presented. Clean this manuscript and follow that standard.  

Missing from the introduction are the research questions/objectives. Because of this gap, the discussion section is not focused. 

Methods:

This section was badly presented. It is full of mathematical details, which one can find in literature. Present the actual steps taken to undertake this study. You need to indicate how many study sites were involved. Explain the selection of the chosen time periods for taking photos and include references there too. Include a reference or motivation for why an iPhone was selected as the tool.  

You must include information on how the diseases were identified in the field and how many photos of each disease were taken.

In line 114, provide details of how the labeling was done. 

You need to provide actual steps done to analyze the photos/images and not only the mathematical details about this process. 

Experimental details or the design were not presented at all. It is bad science to just throw in terms or equations without presenting what they mean. Clarify to the reader what target frames, a target box, and a target center is.

Please note that mathematical equations or details should not replace method steps in the manuscript. 

Results section:

This section is mixed with method details. It needs to be cleaned. 

The section is full of subjective judgments and not factual reporting of findings. In lines 273-278. indicate how judged or how determined.

The ablation experiments as well as the performance Comparison of different models must be presented/explained in the method section and not in the results section.

Discussion section:

The first part of the discussion is not focused. In it, readers want to know what was the aim of the study and brief a summary of what was found (the first 200 or so words).

The discussion section is not comprehensive; it should clearly establish the link between your introduction, the aims/objectives, and your results. In other words, it should answer objectively the question of whether the proposed method is superior and in what way to previous methods. 

The authors need to clearly:

* Assess their findings in light of their objectives/questions.

* They should tell readers whether findings contradict, improve or support what has been found/published by others doing similar work.

* Then provide a take-home message from that assessment.

All subjective judgments should be removed from the manuscript

Although dealing with an interesting topic, the manuscript is poorly presented. 

Important experimental details and method steps are missing. Instead, the authors present a lot of mathematical detail which readers can find for themselves in the literature.  It should not be published in this state.

It is not clear how this method is better than the others already published and the steps of how this aspect is judged should be presented. 

Author Response

We thank you for taking the precious time to read our manuscript. Regarding your corrections, I have revised them and please check the attachment for details.

Reviewer 2 Report

1) In the Abstract, please mention what does FCOS-FL Stand for?

2) In most places, only abbreviation is used. If possible, include an abbreviation table.

3) Include the novelty of the work pointwise in the introduction section. 

4) In the data collection subsection, it is mentioned, "To simulate the actual fruit collection process, the shooting angle was set to be 136

vertical 90°, upward 45°, and downward 45°" Why this shooting angle were used explain. 

5) Kindly check the name in  Section 2.5 Model Evalua cators, 

6) All the graph and figure quality needs to be enhanced. Also, kindly provide the citation of Figures 3, 4, 5, and 6. 

Author Response

(The authors gave the same response as above.)

Reviewer 3 Report

The article deals with the detection of leaf diseases and insect pests using a specific target detection technique that has been optimized in this work for this application - improved fully convolutional one-stage object detection (FCOS).

Generally speaking, the subject is well presented, and methodology and achievement are described adequately. The presented work will appeal to the readers of Agronomy and may be accepted for publication after minor revisions.

Some suggestions:

1.       When first used, the term FCOS  should be introduced as Fully Convolutional One-Stage Object Detection (FCOS).

2.       Line 114 …" The labelImg tool was used to label the 3725 Litchi disease and pest images mentioned above"…. What is labelImg? Who created it? Labeling is time-consuming; the authors might want to discuss  https://doi.org/10.1016/j.compag.2023.107709 for a higher throughput labeling process. How does the  Rooster software (https://github.com/12HuYang/Rooster) compare?

3.       What platform did the authors use? Python? Matlab? Something else?

4.       Graphics should be improved in all figures. For example, the word feature on the right hand of Figure 4 is truncated.

5.       The dataset is relatively small, but it seems to be highly diverse. Yet, there is no attempt to test the method's generalizability. For example, say we train using samples taken during spring and summer time; which set of parameters will be best for samples taken during the autumn? In other words, testing for accuracy is essential, but it is not the only important parameter to consider. Obviously, the method is optimized for accuracy, but the reader will be interested in its robustness. The authors might want to add some calculations to the paper or at least consider this issue theoretically in the current work. 1.       They may want to use for this discussion regarding the method's generalizability,  the following reference, or any other suitable references: https://doi. org/10.3389/fpls.2019.00941, https://doi.org/10.1016/j. inpa.2019.11.001, https://doi.org/10.1016/j.compag.2022.106732 , https://doi.org/10.3390/s21051601.

Author Response

(The authors gave the same response as above.)

Round 2

Reviewer 1 Report

The manuscript has matured in a big way.  It now requires cleaning up and removing repeated statements. Some sentences are too long and difficult to understand. The authors need to shorten them. 

The Litchi leaf diseases and pest problems can cause a decrease in Litchi production, a decline in fruit quality, and a reduction in income for farmers. In order to explore a method for accurately identifying Litchi leaf diseases and pests in real-time and to minimize the harm of pests and diseases to Litchi plants, field surveys were conducted in three different orchards to investigate
and summarize the types, degrees, characteristics, and control measures of Litchi leaf diseases and pests. Eventually, five common Litchi leaf diseases and pests (Litchi leaf mite, Litchi sooty mold, Litchi anthracnose, Mayetiola sp., and Litchi algal spot) were identified as research objects, and Litchi
image data infected with these pests and diseases were used as research targets.

The above sentence is too long and convoluted. The meaning is unclear. Revise this part using shorter sentences.

You must make the statement concise by deleting the words starting from “….summarize the types, degrees, characteristics, and control measures of Litchi leaf diseases and pests”. Furthermore, you must delete the sentence about method details from this point, lines 22-24) because these same words or method details are explained again in lines 26-29

The CMPA (central moment pooling attention) attention mechanism is introduced to enhance the characteristics
of Litchi disease.
In my previous comments, I indicated that the definition, full name, or description should come first and then the abbreviation should come last in brackets. Fix this issue here as well (lines 25-26).  

Lines 28-29 “
An improved location loss function is proposed, which adds difficultly divided sample weights to the FCOS training process to improve the location accuracy of the model” This is a repetition of lines 22-24. Rephrase

Introduction:

In the section covered by lines 106-115, although the objectives are now well-presented, the grammar is not correct. Clean these sentences upFor example, “To address the issues of low accuracy and large model parameters in the current methods for detecting Litchi leaf diseases and pests, and to assist in the prevention and yield increase of Litchi diseases and pests…... [Rephrase this line, particularly clarifying what is meant byassisting in the prevention and yield increase of Litchi disease and pest”. Meaning is obscure.

Materials and methods:

Line 356- 357, Include footnotes to assist readers with interpreting Table 4, Table 5, and Table 6

Discussion:

The discussion section is still not comprehensive; it should clearly establish the link between your introduction the objectives and your results. In other words, it should answer objectively the question of whether the proposed method is superior and in what way to previous methods. It should also provide the limitations of this study. Furthermore, the authors need to:

  • Tell readers whether findings contradict or support what has been found/published by others doing similar work.

Reduce the lengths of your sentences. They are difficult to understand in places

Author Response

(The authors gave the same response as above.)
